# Normal References of Peak Oxygen Uptake for Cardiorespiratory Fitness Measured with Cardiopulmonary Exercise Testing in Chinese Adults

**DOI:** 10.3390/jcm11164904

**Published:** 2022-08-21

**Authors:** Yan Wang, Huijuan Li, Juan Wang, Wei Zhao, Zhipeng Zeng, Li Hao, Yifang Yuan, Yuwei Lin, Yangfeng Wu, Zhengzhen Wang

**Affiliations:** 1School of Sports Medicine and Rehabilitation, Beijing Sport University, 48 Xinxi Road, Haidian District, Beijing 100084, China; 2Peking University Clinical Research Institute, Peking University Health Science Center, 38 Xueyuan Road, Haidian District, Beijing 100191, China; 3Peking University Third Hospital, 49 Huayuan North Road, Haidian District, Beijing 100084, China

**Keywords:** cardiorespiratory fitness, cardiopulmonary exercise testing, peak oxygen uptake, Chinese adults, normal reference values

## Abstract

Introduction: This study aims to establish normal reference values of peak oxygen uptake (VO_2peak_) for cardiorespiratory fitness (CRF) in Chinese adults using cardiorespiratory exercise testing (CPET). Methods: A cross-sectional study was done in four communities, two in the North (Beijing) and two in the South (Hezhou, Guangxi) of China from 1 January 2017 to 31 December 2018, with one urban and one rural in each region. Out of 1642 participants screened, 1114 were eligible and completed CPET using a cycle ergometer (Ergosana320F) without abnormal ECG and were included in the analysis. The 2nd and 98th percentiles of $\overset{\cdot }{V}$O_2peak_ were used as the lower and upper limits of the normal reference values. Results: Significant difference in mean
$\overset{\cdot }{V}$O_2peak_ was shown between men (27.0 mL·min^−1^·kg^−1^) and women (23.7 mL·min^−1^·kg^−1^). The mean $\overset{\cdot }{V}$O_2peak_ decreased with age in both sexes, from 35.8 mL·min^−1^·kg^−1^ in age 20–29 years to 20.5 mL·min^−1^·kg^−1^ in 70–79 years in men and from 29.2 mL·min^−1^·kg^−1^ to 17.0 mL·min^−1^·kg^−1^ in women. Thus, the age- and sex-specific normal reference values of $\overset{\cdot }{V}$O_2peak_ were presented for each 10-year age group by men and women separately. Conclusions: This first community-based study in China provides age- and sex-specific normal references of $\overset{\cdot }{V}$O_2peak_ as a measure of CRF in Chinese adults, which differed significantly from those established in Western populations. Future studies with national representative samples should be warranted.

## 1. Introduction

Cardiorespiratory fitness (CRF) reflects the comprehensive cardiac and pulmonary function and the ability of muscles to use oxygen. Reduced CRF was found strongly associated with the risk of cardiovascular death and all-cause mortality [1,2,3,4]. For the general population, CRF may help to improve the accuracy of the individualized patient risk assessment and guide clinical decisions [5]. Cardiopulmonary exercise testing (CPET) using respiratory gas exchange analysis during incremental exercise remains the gold standard approach for the assessment of an individual’s exercise capacity [6]. Among the multiple measures for CRF, though $\overset{\cdot }{V}$O_2max_ produced by progressive maximal exercise may not be assured with an exercise-naïve or unmotivated population, and peak oxygen uptake ($\overset{\cdot }{V}$O_2peak_) is simply the highest $\overset{\cdot }{V}$O_2_ reached in the given test, $\overset{\cdot }{V}$O_2peak_ is still mostly used due to the convenience for measurement [7,8,9]. However, most of the current studies on CRF were from high-income countries, and data from low-and middle-income countries are still lacking. A systematic review published in 2019 included 78 CRF studies of the general population, of which only 2 were conducted in low- and middle-income countries [10]. Another systematic review on normative CRF standards included 29 studies, and among those, 16 were from Europe, 9 from North America, and 5 from South America, and CRF data in a general Asian population were not available [11]. CRF may have racial differences and may be affected by social economic status [12]; thus, conducting CRF researches in low-and middle-income countries is urgently needed [11].

In China, the Healthy China Initiative was launched in 2016, and one of its goals is to increase the number of people who regularly participate in physical exercise from 360 million in 2015 to 530 million in 2030 [13]. However, the safety concerns for exercise are a real hurdle to the mass physical exercise endeavor. To help Chinese people exercise safely, reference values of CRF based on healthy Chinese individuals from the general population are also urgently needed, though a recent study of $\overset{\cdot }{V}$O_2peak_ reference values was reported using the hospital electronic medical record data of Chinese individuals without specific clinical complaints [14]. The importance of research on normal references of CRF based on healthy individuals from general population has been emphasized by the American Heart Association (AHA) in 2013 [15]. Here, we aimed to establish the standard references of $\overset{\cdot }{V}$O_2peak_ with data from a cross-sectional survey of healthy Chinese adults screened out from four typical urban and rural residential communities in northern and southern China.

## 2. Methods

### 2.1. Study Design and Participants

For the purposes of the study, a cross-sectional study was conducted among four samples of apparently healthy Chinese adults through a screening of residential communities in Hezhou and Guangxi (South) and Beijing (North) of China from 1 January 2017 to 31 December 2018. In each city, one urban and one rural community were chosen for the screening to recruit the eligible participants for the CPET. The screening employed a questionnaire on the history of diseases and lifestyles in addition to measuring the blood pressure and pulse rate of every participants. Eligible participants were men and women aged 20–79 years with no obvious movement disorders. Exclusion criteria included: (1) major cardiovascular disease; (2) blood pressure (BP) ≥ 160/100 mmHg; (3) neuromuscular and skeletal muscle diseases; (4) acute systemic infection with fever, pain, or lymphadenopathy; (5) mental or physical disorders that lead to significantly movement disorders; and (6) other diseases that are not suitable for exercise. The questionnaire interview and CPET were conducted by trained staff under close guidance and using the same protocol. The goal of recruitment was to have approximately 25 men and 25 women in each age group: 20–29 years, 30–39 years, 40–49 years, 50–59 years, 60–69 years, and 70–79 years. There would be 1200 participants in total, half men and half women, half urban and half rural. All study participants were provided with detailed information, and data collection started after the informed consent was completed. The study was reviewed and approved by the Beijing Sport University Institution Review Board (#2016022H).

### 2.2. Data Collection

The basic demographic information and history of disease were collected with the questionnaire. Height and body weight were measured with standard protocol [16]. Seated blood pressure was taken on right upper arm using automatic blood pressure monitor (OMRON, Model HEM-7124) for three times, with at least 30 s intervals.

### 2.3. Methods of Cardiopulmonary Exercise Testing

The exercise testing was carried out with a cycle ergometer (Ergosana320F; Ergosana, Bitz, Germany) using a graded exercise test (GXT) protocol, expired gas (cosmed, software: Onimia), and ECG measurement (DMS 300-BTT01). First, the participants were allowed to adaptively pedal for 1 min without any load; then, in the formal test, the starting load was 25 W, the speed was 50 r/min, and the increment was 25 W every 3 min. ECG monitoring was used throughout the test. The blood pressure and heart rate were recorded before the end of each level of workload. Meanwhile, the Borg Rating of Perceived Exertion (RPE) scale was used to describe the intensity level participants were experiencing every 10 s before the end of each workload. When the participants reached one of the following termination criteria, the workload was returned to zero; and the duration time, heart rate, RPE, and blood pressure were recorded immediately. Then participants were requested to stay still for five minutes after the test, during which heart rate and blood pressure were continually measured once per minute.

### 2.4. Criteria for Termination of the Test

Termination criteria were developed partially referring the ACSM’s Guidelines for Exercise Testing and Prescription (10th) [17]. Participants with any of the following conditions should terminate the test: (1) reaching or exceeding 85% of the maximum predicted heart rate (207—age × 0.7) or 70% of the reserve heart rate ((maximum heart rate—rest heart rate) × 70% + rest heart rate); (2) RPE ≥ 17; (3) respiratory exchange ratio (RER) ≥ 1.1; (4) systolic blood pressure ≥ 230 mmHg at any time during exercise; (5) abnormal ECG; (6) heart rate not rising or falling with the increase of exercise load; (7) showing signs of myocardial ischemia, hypoperfusion, central nervous system symptoms, dyspnea, abnormal fatigue, and muscle spasm or other symptoms or signs; or (8) subjects requesting to stop. The modification was done to increase the safety of the test that was conducted at the community centers without on-site medical staff support.

### 2.5. Statistical Analysis

Our study followed the STROBE cross sectional reporting guidelines [18]. Participant characteristics were described by means (SD) for continuous variables and frequency for categorical variables. Mean $\overset{\cdot }{V}$O_2peak_ (mL·min^−1^·kg^−1^) were described by age and sex groups. Age- and sex-specific centile curves were produced using generalized additive models adjusting for location, scale, and shape (GAMLSS package, R version 4.0.2). Percentile distribution was used to set the reference range’s upper and lower limits at 98th and 2nd percentile. The age trajectories were modeled using P-splines. Lambda-Mu-Sigma (LMS) and its extensions were used to fit the centile curves.

## 3. Results

### 3.1. Demographic Information

A total of 1642 participants were screened in the study. Among all the potential participants, 423 who met the exclusion criteria were screened out. In the remaining 1193 eligible participants, 8 did not complete the CPET due to machine failure, and 71 had positive or suspected positive ECG abnormal findings during the exercise test. Thus, our analysis included 1114 participants who completed CPET with normal ECG during exercise test (Figure 1). Age and sex distribution and other demographic characteristics are displayed in Table 1. As expected, the number of participants in the age group of 70 and older was smaller.

### 3.2. CPET Parameters

Among 1114 participants who completed the CETP test, 86.7% reached at least one of the criteria for stopping at the time when the test was terminated (Table 2). Interestingly, more women than men had a heart rate ≥ 85% of the maximum predicted heart rate, HRR ≥ 70%, and RER ≥ 1.1. On the contrary, more men than women had a RPE score ≥ 17 and the highest SBP ≥ 230 mmHg. In particular, the number of men who had the highest SBP ≥ 230 mmHg was three-fold that in women. CPET responses at maximal effort in both sexes are presented in Table 3.

The average $\overset{\cdot }{V}$O_2peak_ was 27.0 mL·min^−1^·kg^−1^ in men and 23.7 mL·min^−1^·kg^−1^ in women, and it was higher in men than in women at every age group, showing a downward trend with age in both sexes (Table 4 and Figure 2).

## 4. Discussion

Our study first reported normal reference values of $\overset{\cdot }{V}$O_2peak_ on the basis of data from community-based apparently healthy Chinese adults in urban and rural areas of the north and south of China. The strict inclusion and exclusion criteria, standard test methods and procedures, and the same team for field work in all samples guaranteed the quality of the study.

### 4.1. Comparison of Our Findings with Previous Studies

Consistent with findings from previous studies in other countries [11], our study also found that $\overset{\cdot }{V}$O_2peak_ was higher in men than that in women, and it reduced with advancing age. However, the sex difference in $\overset{\cdot }{V}$O_2peak_ in our study was about 13%, which is smaller than that reported in the FRIEND Registry (about 27%) and that in a systematic review conducted in 2019 (about 20%) [11,15,19]. Meanwhile, we found that the $\overset{\cdot }{V}$O_2peak_ in Chinese men declined much faster in the age range from 20 to 30 years than in the later age groups (Figure 2), and this phenomenon was not reported in previous studied in Americans and Europeans [19].

Compared with the results of the FRIEND Registry in the United States, mean $\overset{\cdot }{V}$O_2peak_ in our study was lower in all age groups. The largest difference was in the age group of 20–29 years, which was 14 mL·min^−1^·kg^−1^ in men and 10 mL·min^−1^·kg^−1^ in women, while the difference in age group of 70 to 79 years was the smallest: 4 mL·min^−1^·kg^−1^ for men and 1 mL·min^−1^·kg^−1^ for women. In addition to race, height, and weight, the main reasons explaining the large discrepancies in $\overset{\cdot }{V}$O_2peak_ may include the following. First, our research was conducted in non-medical environment, and in order to ensure participants’ safety, the criteria for exercise test termination was stricter. In addition to respiratory quotient, Borg RPE for each stage (6–20) and 85% of the maximum age-predicted heart rate for termination were also used in our tests. Second, we used a cycle ergometer, but the FRIEND Registry used a treadmill. Previous studies showed that $\overset{\cdot }{V}$O_2peak_ is typically 5–20% lower during a maximal exercise test performed on a cycle ergometer compared to a treadmill, and a 10% difference is typically used by clinicians when comparing peak exercise responses between cycle ergometry and treadmill exercise [20,21,22]. The lower CRF of the Chinese population in comparison with that of European and American populations might put them at a higher risk of functional limitation or disability if the same level of CRF is associated with the same level of the risks in different ethnic groups. However, the current normal references are based on a cross-sectional analysis of the CRF indices from apparently healthy populations. Without sufficient data from longitudinal cohort follow-up studies that allow to establish and compare the association between CRF and the risk of mortality and morbidity between ethnic groups, we are not sure that is true or not.

Mean $\overset{\cdot }{V}$O_2peak_ values reported in our study was higher compared with that reported by Dun and colleagues in 2021 among Chinese patients who had no symptoms recorded. For example, for people aged from 20 to 30 years in both genders, mean $\overset{\cdot }{V}$O_2peak_ values were about 30% higher in our study [14]. We believe this discrepancy should be mainly due to the difference in source of the study samples. Our study samples were community-based, while Dun’s study was based on electronic hospital records. Although Dun and colleagues carefully selected those asymptomatic ones, patients seeking medical advices at a hospital cannot be as healthy as people living freely in the communities. Our study was pre-designed and used the same device, staffs, and standard operation procedures for the CPET, while Dun’s analyzed retrospective data from an over-10-year span, where the device, staff, and operation procedures might all change.

### 4.2. The Possible Impact of Stricter Termination Criteria in CPET

Our research tentatively used stricter criteria for terminating the CPET to ensure the safety of the participants, as excessive exercise intensity is the main factor inducing cardiovascular events during exercise [23]. Though some studies discussed the low probability of safety incidents during high-intensity exercise [24,25], the design of our study is relatively conservative considering a community-based setting. The termination principle of our research prioritized the use of 70% heart rate reserved and RPE. Thus, the normal reference standards reported in our study should not be extended to the hospital settings where sufficient medical resources are available to ensure the patients’ safety when any accidental events take place. Although it would be safer for the participants, it would probably lead to a lower level of CRF and could be a reason why our results are lower than those reported from the Western population. However, our results are still higher than that reported by Dun and colleagues using hospital electronic medical record data of Chinese individuals without specific clinical complaints [11], indicating that the source of study population (community-based or hospital-based) may be much more important than the stopping criteria. The advantage in developing the CRF reference standards with the methods in our study is that it maximizes the safety when applying it in mass physical exercises where medical resources are not sufficiently available. Thus, the CRF reference standards developed from our study should be more practical and applicable to the Healthy China 2030 initiative, which calls for nationwide mass physical fitness efforts with improved physical fitness monitoring systems and carrying out sports risk assessment [13,26].

There are several limitations in our study. First, our study samples were drawn from only two regions of China due to funding limits. Although we tried our best to recruit participants from the north and south of China, the national representativeness was not warranted. Second, we used stricter criteria to terminate the CPET, and hence, the reference standards developed in our study might be lower than that obtained from CPET in hospitals or other settings where medical resources are adequate. However, as discussed above, it has pros and cons. Third, our screening for eligibility was mainly by interview and did not employee any physical examinations except for blood pressure measurement. Thus, our study participants can only be claimed as apparently healthy. However, we believe the use of more physical examinations, particularly those with medical equipment, should not only increase expenses but more importantly limit the generalizability of the reference standards.

## 5. Conclusions

This study provides age- and sex-specific normal references of $\overset{\cdot }{V}$O_2peak_ as the measure of CRF in Chinese adults, which differed significantly from that established in Western populations. Future studies with nationally representative samples should be warranted.

## Figures and Tables

**Figure 1 jcm-11-04904-f001:**
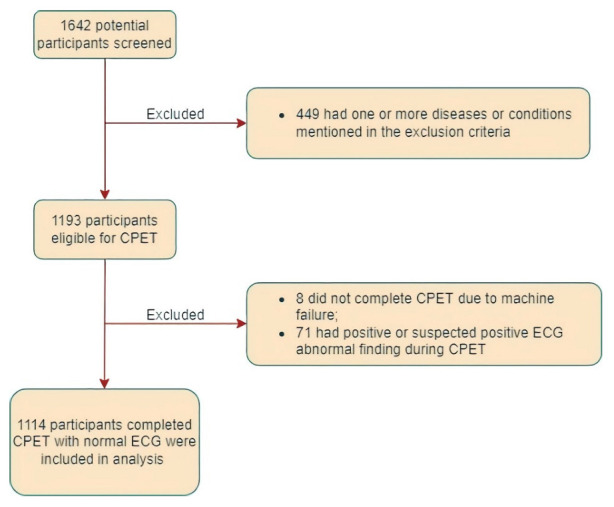
The flowchart for inclusion and exclusion of participants in the study.

**Figure 2 jcm-11-04904-f002:**
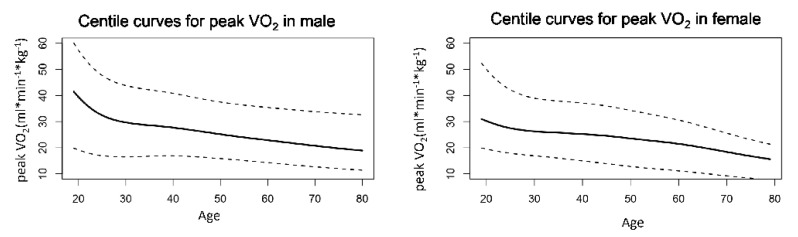
Centile curves of $\overset{\cdot }{V}$O_2peak_ (mL·min^−1^·kg^−1^) for men and women. The dashed lines on the top represent the upper limits at 98th, the solid line in the middle represents the median $\overset{\cdot }{V}$O_2peak_ and the dashed lines at the bottom represent the lower limits at 2nd percentile.

**Table 1 jcm-11-04904-t001:** Demographic characteristics in the Chinese adult population.

	Male	Female
*n* (%)	495 (44.4)	619 (55.6)
Age (%)		
20–29	17.8	14.9
30–39	18.0	17.1
40–49	15.6	18.6
50–59	20.8	19.7
60–69	18.4	18.7
70–79	9.5	11.0
Marriage status (%)		
Single	19.8	17.0
Married	80.2	83.0
Education (%)		
Primary school or less	11.3	19.1
Junior high school	22.0	19.8
High school	22.0	23.5
College and above	44.7	37.6
Smoking (%)	40.2	1.1
Regular exercise habits (%)		
>30 min/day, 3 days/week	30.1	33.8
Height in cm (mean ± SD)	166.7 ± 6.7	156.0 ± 6.0
Weight in kg (mean ± SD)	68.0 ± 11.3	57.4 ± 9.2
BMI (%) *		
<18.5 kg/m^2^	3.2	4.3
18.5~23.9 kg/m^2^	44.2	55.1
24~27.9 kg/m^2^	36.4	30.7
28~30 kg/m^2^	16.2	9.9
Blood pressure (%)		
SBP < 140 and DBP < 90 mmHg *	71.3	81.7
SBP ≥ 140/DBP ≥ 90 mmHg	28.7	18.3

* BMI, body mass index; SBP, systolic blood pressure; DBP, diastolic blood pressure.

**Table 2 jcm-11-04904-t002:** Subjects who reached the termination criteria (by number and percent).

Termination Criteria	Men	Women	Total
*n*	495	619	1114
HR_max_/predictHR_max_ ≥ 85%	289 (58.4%)	437 (70.6%)	726 (65.2%)
HRR ≥ 70%	312 (63.0%)	469 (75.8%)	781 (70.1%)
RER ≥ 1.1	85 (17.2%)	151 (24.4%)	236 (21.2%)
RPE ≥ 17	242 (48.9%)	289 (46.7%)	531 (47.7%)
Highest SBP ≥ 230	79 (16.0%)	29 (4.7%)	108 (9.7%)
Abnormal ECG	2 (0.4%)	9 (1.5%)	11 (1.0%)
Increased exercise load with no heart rate rise or fall	0 (0.0%)	1 (0.2%)	1 (0.1%)
Signs of symptoms *	4 (0.8%)	5 (0.8%)	9 (0.8%)
Subject request to stop	68 (13.7%)	59 (9.5%)	127 (11.4%)

* Signs of symptoms include: myocardial ischemia, hypoperfusion, central nervous system symptoms, dyspnea, abnormal fatigue, and muscle spasm or other symptoms or signs.

**Table 3 jcm-11-04904-t003:** CPET responses at maximal effort in men and women.

	Men	Women	Total
RPE *	15.9 ± 1.9	15.8 ± 1.7	15.8 ± 1.8
HR_peak_ *	150.1 ± 20.8	154.3 ± 18.1	152.3 ± 19.4
HR_peak_/predictHR_max_ *	0.86 ± 0.09	0.89 ± 0.08	0.88 ± 0.09
HRR% *	74.5 ± 16.3	80.0 ± 14.8	77.4 ± 15.7
RER *	1.00 ± 0.11	1.04 ± 0.10	1.02 ± 0.10
Termination load (Watt)	132.7 ± 32.0	105.9 ± 27.1	117.8 ± 32.2

***** RPE, Rating of Perceived Exertion scale; HRpeak, the peak heart rate during CPET test; HRR, heart rate reserved; RER, respiratory exchange ratio.

**Table 4 jcm-11-04904-t004:** Sex-specific percentiles of measured $\overset{\cdot }{V}$O_2peak_ (mL·min^−1^·kg^−1^).

Age Group (y)	*n*	Mean	SD				Percentile					
				2nd	5th	10th	25th	50th	75th	90th	95th	98th
Man												
20–29	88	35.8	9.7	17.9	19.1	24.1	28.1	35.7	44.9	48.9	50.7	51.9
30–39	88	28.8	6.6	14.3	19.0	20.8	24.3	27.7	33.2	37.9	41.6	43.9
40–49	76	27.4	5.5	18.6	18.9	21.0	23.4	26.7	31.5	34.6	36.7	40.1
50–59	102	24.4	5.1	15.9	17.2	17.6	20.6	24.4	27.1	31.0	33.5	35.5
60–69	89	22.9	5.1	13.9	14.9	15.8	19.3	22.4	26.0	29.4	31.9	34.7
70–79	47	20.5	5.5	11.5	13.5	14.2	15.8	19.7	22.9	29.1	31.1	34.5
Subtotal	490	27.0	8.1	14.2	15.8	17.7	21.5	25.9	31.0	37.7	44.0	48.8
Woman												
20–29	92	29.2	7.0	16.0	20.9	21.8	25.3	27.7	32.8	37.5	41.7	49.8
30–39	105	25.6	4.8	15.7	18.4	19.6	22.1	26.0	28.8	31.8	34.2	34.6
40–49	114	24.9	5.0	12.5	16.7	20.1	22.1	24.7	27.5	30.7	32.7	36.8
50–59	122	22.6	4.7	14.0	16.5	17.8	20.0	22.0	25.2	28.5	31.0	33.8
60–69	114	21.1	4.9	11.1	12.3	13.8	17.7	21.2	24.4	28.0	29.2	30.8
70–79	64	17.0	4.1	8.2	9.5	11.6	14.4	17.0	20.0	21.9	22.8	23.1
Subtotal	611	23.7	6.2	11.4	13.5	16.4	20.1	23.1	27.0	31.1	34.0	37.5

## Data Availability

Data available on request due to ethical restrictions. The data presented in this study are available on request from the corresponding author. The data are not publicly available due to ethical restrictions.

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
