# Peer review of "Normal References of Peak Oxygen Uptake for Cardiorespiratory Fitness Measured with Cardiopulmonary Exercise Testing in Chinese Adults"

_jcm, 2022, doi:10.3390/jcm11164904_

Round 1

Reviewer 1 Report

I completed the revision of the manuscript entitled  “Normal References of Peak Oxygen Uptake for Cardiorespiratory Fitness Measured With Cardiopulmonary Exercise Testing in Chinese adults”.  The manuscript by Yan Wang, et al. describes an environmental study in China that provides normal VO2peak references for age and sex as a measure of CRF in Chinese adults, which, as presented, differ significantly from those established on Western populations. The structure of the manuscript (Introduction, Methodology, Results, Discussion) follows a logical sequence. Introduction contains enough background informations.  The research is well designed and carried out. Materials and methods are clearly described. This is interesting and well-written article.

Specific minor comments:

1.      Line 18 and 20 in the Abstract section: the sentence should start with a capital letter.

2.      Line 289 - in item 22 of the Reference, provide the name of the person by whom the cited communication was written for the American College of Sports Medicine.

Author Response

Point 1: Line 18 and 20 in the Abstract section: the sentence should start with a capital letter.

Response 1: Thank you very much for your appreciation of our work and pointing out our mistakes. We have change those sentences with capital letters.

Point 2: Line 289 - in item 22 of the Reference, provide the name of the person by whom the cited communication was written for the American College of Sports Medicine.

Response 2: Done. The name of the person added in reference 22.

Reviewer 2 Report

The paper by Wang and colleagues want to establish normal reference values of peak oxygen uptake for cardiorespiratory fitness (CRF) in Chinese adults using cardiorespiratory exercise test. I agree with the Authors that data is still limited within this large and heterogeneous population, and their results may help fill the lack of reference values. Although reported among the study limitations, I believe that the use of stricter limitation criteria is the main drawback of the study, because, together with the use of a bike as testing ergometer it allows to probably significantly underestimate the exercise capacity in this population. Despite this, the paper is of interest because of the highlighted data limitation on this topic. In general, I would improve both the Introduction section, by adding a sentence about the importance of having a higher VO2max (i.e. the reduction on cardiovascular and all-cause mortality for each 1-MET increase of exercise capacity, maybe citing studies on a Chinese population, if present). In the Discussion section, I also would add a comment about the potential importance of the lower CRF in the Chinese population respect to others (European, American), in particular for the higher age classes and female subjects: may this put them at a higher risk of functional limitation and disability in their old age? (all this, still considering the limitation of submaximal tests).

I also have a few questions I wish the Authors could answer:

1.     Don’t You think that including within the population also (part of if not all) subjects that terminated the test because of the conditions reported as point 4, 5, 6, 7 may have affected your results? I mean, I don’t know if you can rely on those results as expression of a maximal effort. Please, also motivate why You decided to include those 13.3% of subjects who did not meet at least one of the criteria for exercise termination.

2.     Do You think that exercise testing modalities (i.e. step vs ramp) may have affected the study results?

3.     Since overweight/obesity can affect VO2/kg values, I wonder if the prevalence of these condition in your population is similar to that in the general one.

4.     You affirm that a smaller number of participants in the group age 70 and older was expected: why You didn’t take into account this predictable limitation by recruiting a higher number of this age category (then offsetting the likely larger number of subjects meeting the exclusion criteria).

Minors: please check some superscript characters (i.e. min-1), some spaces and minor spelling mistakes throughout the manuscript.

Author Response

Point 1: The paper by Wang and colleagues want to establish normal reference values of peak oxygen uptake for cardiorespiratory fitness (CRF) in Chinese adults using cardiorespiratory exercise test. I agree with the Authors that data is still limited within this large and heterogeneous population, and their results may help fill the lack of reference values. Although reported among the study limitations, I believe that the use of stricter limitation criteria is the main drawback of the study, because, together with the use of a bike as testing ergometer it allows to probably significantly underestimate the exercise capacity in this population. Despite this, the paper is of interest because of the highlighted data limitation on this topic. In general, I would improve both the Introduction section, by adding a sentence about the importance of having a higher VO2max (i.e. the reduction on cardiovascular and all-cause mortality for each 1-MET increase of exercise capacity, maybe citing studies on a Chinese population, if present). In the Discussion section, I also would add a comment about the potential importance of the lower CRF in the Chinese population respect to others (European, American), in particular for the higher age classes and female subjects: may this put them at a higher risk of functional limitation and disability in their old age? (all this, still considering the limitation of submaximal tests).

Response 1: Thanks for your seeing the value of our study. We will take your suggestions to improve the Introduction and Discussion parts. We also thank you for reminding us to discuss about the potential importance of the lower CRF in the Chinese population respect to others (European, American). Currently, the normal references of CRF test were basically established through a cross-sectional survey of apparent healthy populations rather than longitudinal cohort follow up studies that link baseline CRF level with subsequent risk of mortality and morbidity. Without such longitudinal data compatible between Chinese and other populations, it is hard to understand if a Chinese with the CRF lower than an American should necessarily have a higher health risk. For example, it is widely accepted that a Chinese would develop diabetes and other health problems at a lower level of body mass index. Thus, the Chinese Guidelines on Obesity and Overweight defined overweight as with BMI >=24 Kg/m2 and obesity with BMI >=28 Kg/m2, which were based on analysis of large cohort follow up databases and different from that defined by the WHO.

Manuscript changes:

First, we have added the following sentence into the Introduction. Unfortunately, there is no study in Chinese population on the association of exercise capacity with the risk of cardiovascular disease or total mortality.

“Reduced CRF was found strongly associated with the risk of cardiovascular death and all-cause mortality [Ezzatvar, Yasmin et al. “Cardiorespiratory fitness measured with cardiopulmonary exercise testing and mortality in patients with cardiovascular disease: A systematic review and meta-analysis.” Journal of sport and health science vol. 10,6 (2021): 609-619. doi:10.1016/j.jshs.2021.06.004]”.

Second, we add the following sentence in the Discussion part about the potential importance of a lower CRF in Chinese.

“The lower CRF of Chinese population, in comparison with that of European and American populations, might put them on a higher risk of functional limitation or disability if the same level of CRF is associated with the same level of the risks in different ethnic groups. However, the current normal references are based on cross-sectional analysis of the CRF indices from apparently healthy populations. Without sufficient data from longitudinal cohort follow up studies that allows to establish and compare the association between CRF and the risk of mortality and morbidity between ethnic groups, we are not sure that is true or not.”

Point 2: Don’t You think that including within the population also (part of if not all) subjects that terminated the test because of the conditions reported as point 4, 5, 6, 7 may have affected your results? I mean, I don’t know if you can rely on those results as expression of a maximal effort. Please, also motivate why You decided to include those 13.3% of subjects who did not meet at least one of the criteria for exercise termination.

Response 2: As we discussed in the manuscript, the pre-specified stopping criteria listed out there were to ensure the safety of the test to the participants. These criteria were developed partially referring the ACSM's Guidelines for Exercise Testing and Prescription (10th). Table 2 listed only four of the seven criteria for stopping the test, the 13.3% of participants who were included in the study stopped the test for other reasons: 1.0% due to abnormal ECG during exercise, 0.8% due to showing signs of myocardial ischemia, hypo perfusion, central nervous system symptoms, dyspnea, abnormal fatigue and muscle spasm, or other symptoms or signs, 0.1% due to increased exercised load with no heart rate rise or fall and 11.4% stopped due to subjects’ request. Comparing to the ACSM’s Guidelines, our criteria were stricter. Although it would be safer for the participants but it would probably lead to a lower level of CRF and could be a reason why our results are lower than those reported from Western population. However, our results are still higher than that reported by Dun and colleagues using hospital electronic medical record data of Chinese individuals without specific clinical complaints, indicating the source of study population (community-based or hospital-based) may be much more important than the stopping criteria.

Manuscript changes: We add “subjects request to stop” as one of the test termination conditions and the number of participants included for other reasons in Table 2. We also add the following sentence at the end of the paragraph describing the stopping criteria.

“The modification was done to increase the safety of the test that was conducted at the community centers without on-site medical staff support.”

We add in the Discussion the following sentence to discuss the possible impact of using a stricter stopping criteria.

“Although it would be safer for the participants but it would probably lead to a lower level of CRF and could be a reason why our results are lower than those reported from Western population. However, our results are still higher than that reported by Dun and colleagues using hospital electronic medical record data of Chinese individuals without specific clinical complaints[11], indicating the source of study population (community-based or hospital-based) may be much more important than the stopping criteria.”

Reference: American College of Sports Medicine. ACSM's Guidelines for Exercise Testing and Prescription (10th). Lippincott Williams & Wilkins.2017.

Point 3: Do You think that exercise testing modalities (i.e. step vs ramp) may have affected the study results?

Response 3: Thank you very much for your question. According to the literature, for ramp vs step-incremental cycle ergometer tests, most studies have shown no significant difference for the VO2peak and HRpeak despite the ramp protocol having greater peak power output. However, the mean values for VO2 and HR were lower during the ramp incremental test than during the step test with the same workload increments at submaximal intensity. Few literatures reckoned that Ramp is higher than Step. Step-incremental cycle ergometer test was used in our study.

Reference:

  • Caen K , Pogliaghi S , Lievens M , et al. Ramp vs. step tests: valid alternatives to determine the maximal lactate steady-state intensity?[J]. European Journal of Applied Physiology, 2021.
  • Zuniga J M , Housh T J , Camic C L , et al. Neuromuscular and metabolic comparisons between ramp and step incremental cycle ergometer tests[J]. Muscle & Nerve, 2013, 47(4):555-560.

[3] Kamil MICHALIK , Natalia DANEK, Marek ZATOŃ,et al.Assessment of the physical fitness of road cyclists in the step and ramp protocols of the incremental test [J]The Journal of Sports Medicine and Physical Fitness ,2019,59(8):1285-91.

  • Zuniga JM, Housh TJ, Camic CL, et al. Metabolic parameters for ramp versus step incremental cycle ergometer tests. Appl Physiol Nutr Metab. 2012;37(6):1110-1117.

Manuscript changes: None.

Point 4: Since overweight/obesity can affect VO2/kg values, I wonder if the prevalence of these condition in your population is similar to that in the general one.

Response 4: Thanks. We checked the overall distribution of BMI in our subjects and the general population in China. The mean(95%CI) BMI of male and female in our study are 24.4(24.1,24.8) kg/m2 and 23.5(23.3,23.8) kg/m2, respectively; for general population, the mean(95%CI) BMI for male and female are 24.7(24.6,24.9) kg/m2 and 24.1(24.0,24.2) kg/m2 in 2018. (Reference: Wang L, Zhou B, Zhao Z, et al. Body-mass index and obesity in urban and rural China: findings from consecutive nationally representative surveys during 2004–18. The Lancet (British edition). 2021; 398:53-63.) Thus, we considered the prevalence of obesity and overweight similar between our study and the general adult population of China.

Manuscript changes: None.

Point 5: You affirm that a smaller number of participants in the group age 70 and older was expected: why You didn’t take into account this predictable limitation by recruiting a higher number of this age category (then offsetting the likely larger number of subjects meeting the exclusion criteria).

Response 5: Thanks. Although we had expected the smaller number for the age 70- group and had selected the large communities for the recruitment, we did not expected that the response rate was much lower in this senior group than other younger groups. We had contacted all potential eligible older people but failed to recruit the number we planned for age 70 years and above.

Manuscript changes: None.

Point 6: Minors: please check some superscript characters (i.e. min-1), some spaces and minor spelling mistakes throughout the manuscript.

Response 6: Sorry.

Manuscript changes: We went through the whole manuscript to check and correct all possible mistakes.

Reviewer 3 Report

GENERAL COMMENTS

This paper provides a potentially valuable reference fitness chart for Chinese populations with respect to VO2peak.  However, there are multiple concerns that need to be addressed suitably and which detract from the work as currently presented. Specifically:

1.       Some description of the factors limiting VO2max would provide relevant and useful mechanistic information for the reader. Wagner, Exerc Sport Sci Rev. 2000 Jan;28(1):10-4.

2.       The reader needs to be informed of how the methods differ from the state-of-the art VO2max measurement.  See JAPPL Cores of Reproducibility in Physiology VO2max paper: Poole and Jones, JAPPL 2017 Apr 1;122(4):997-1002.

3.       The so-called “secondary criteria” to assess exercise intensity reached have been heavily criticized and this needs perspective.  See Poole et al. EJAP 2008 Mar;102(4):403-10.

4.       The discussion regarding safety is possibly overstated.  Please see reviews by Smith et al. Clin Geriatr Med. 2019 Nov;35(4):469-487 and Rognmo et al. Circulation 2012, 126:1436–1440 which are more current than some cited herein.

5.       The recent work of Rose et al. Exp Physiol. 2022 Aug;107(8):787-799 battled some of the same challenges as herein as regards the CPET and intensity of exercise that they could achieve and, though in patient populations, might provide a useful comparison.

6.       Information on exercise history and smoking are germane and would be useful in Table 1.

7.       Lines 199-.  Normalizing to ml/kg/min for VO2 removes the effects of body size differences one would imagine.

8.       Repeat tests on each patient would have been a great strength.

Specific/Minor

19           Throughout.  It is convention to have a “.” Over the V in VO2.

25           “between men and women”

26           Delete “increasing”

28           Delete “correspondingly”

41           “..the general population”

57           “on” better as “for”

58           “champion” better as “endeavor”

109-16   See Poole et al. 2008 comment above.

Figure 2 needs VO2 units

157         “Quantile” should be “centile”?

197         Delete “was”

199         “Chinese are…”

216         “champion” better as “efforts”

Author Response

Point 1: Some description of the factors limiting VO2max would provide relevant and useful mechanistic information for the reader. Wagner, Exerc Sport Sci Rev. 2000 Jan;28(1):10-4.

Response 1: Thanks.

Manuscript changes: We add the reference in introduction.

Point 2: The reader needs to be informed of how the methods differ from the state-of-the art VO2max measurement.  See JAPPL Cores of Reproducibility in Physiology VO2max paper: Poole and Jones, JAPPL 2017 Apr 1;122(4):997-1002.

Response 2: Thanks.

Manuscript changes: We add the description and reference in introduction.

Point 3: The so-called “secondary criteria” to assess exercise intensity reached have been heavily criticized and this needs perspective. See Poole et al. EJAP 2008 Mar;102(4):403-10.

Response 3: Thanks. We agree that “sencondary criteria” are flawed, yet after comprehensive condideration, our study used termination criteria refered to ACSM's Guidelines for Exercise Testing and Prescription.

Manuscript changes: We add description and reference in introduction.

Point 4: The discussion regarding safety is possibly overstated.  Please see reviews by Smith et al. Clin Geriatr Med. 2019 Nov;35(4):469-487 and Rognmo et al. Circulation 2012, 126:1436–1440 which are more current than some cited herein.

Response 4: Thanks.

Manuscript changes: We have disscussed the limitation of the use of criteria and add the reference in discussion.

Point 5: The recent work of Rose et al. Exp Physiol. 2022 Aug;107(8):787-799 battled some of the same challenges as herein as regards the CPET and intensity of exercise that they could achieve and, though in patient populations, might provide a useful comparison.

Response 5: Thanks.

Manuscript changes: We have disscussed the limitation of the use of criteria and add the reference in discussion.

Point 6: Information on exercise history and smoking are germane and would be useful in Table 1.

Response 6: Thanks.

Manuscript changes: We have added exercise history and smoking status in Table 1.

Point 7: Lines 199-. Normalizing to ml/kg/min for VO2 removes the effects of body size differences one would imagine.

Response 7: Thanks.

Manuscript changes: We removed the relevant sentences.

Point 8: Repeat tests on each patient would have been a great strength.

Response 8: Thanks, we absoulutely agree that repeated measurements are benefical. However, our study had a relatatively large sample size, with over a thound subjects, performing repeated tests are difficult considering resources use and subjects’compliance.

Manuscript changes: None.

MINIORS

Point 9: Throughout. It is convention to have a “.” Over the V in VO2.

Response 9: Thanks. We have changed VO2peak throughoutly.

Point 10: Line 25 “between men and women”

Response 10: Done

Point 11: Line 26 Delete “increasing”

Response 11: Done

Point 12: Line 28 Delete “correspondingly”

Response 12: Done

Point 13: Line 41 the general population

Response 13: Done

Point 14: Line 57 on better as “for”

Response 14: Done

Point 15: Line 58 champion better as “endeavor”

Response 15: Done

Point 16: Figure 2 needs VO2 units

Response 16: Done

Point 17: Line 157 quantile should be “centile”

Response 17: Done

Point 18: Line 197 Delete “was”

Response 18: Done

Point 19: Line 199 “Chinese are…”

Response 19: Done

Point 20: Line 216 champion better as “effort”

Response 20: Done

Round 2

Reviewer 2 Report

I believe that the Authors adequately addressed my requests of revision/clarification.